# Fire Risk Assessment on Wildland–Urban Interface and Adjoined Urban Areas: Estimation Vegetation Ignitability by Artificial Neural Network

**Maria Mahamed (Polinova) [1], Lea Wittenberg [2], Haim Kutiel [3] and Anna Brook [1,*]**

[1] Spectroscopy & Remote Sensing Laboratory, Spatial Analysis Research Center (UHCSISR), Department of Geography and Environmental Studies, University of Haifa, Mount Carmel, Haifa 3498838, Israel
[2] Geomorphology Laboratory, Department of Geography and Environmental Studies, University of Haifa, Mount Carmel, Haifa 3498838, Israel
[3] Climatology Laboratory, Department of Geography and Environmental Studies, University of Haifa, Mount Carmel, Haifa 3498838, Israel
* Correspondence: abrook@geo.haifa.ac.il

**Abstract:** Fire risk assessment on the wildland–urban interface (WUI) and adjoined urban areas is crucial to prevent human losses and structural damages. One of many interacting and dynamic factors influencing the structure and function of fire-prone ecosystems is vegetation ignitability, which plays a significant role in spreading fire. This study sought to identify areas with a high-level probability of ignition from time series multispectral images by designing a pattern recognition neural network (PRNN). The temporal behavior of six vegetation indices (VIs) before the considered wildfire event provided the input data for the PRNN. In total, we tested eight combinations of inputs for PRNN: the temporal behavior of each chosen VI, the temporal behavior of all indices together, and the values of VIs at specific dates selected based on factor analysis. The reference output data for training was a map of areas ignited in the wildfire. Among the considered inputs, the MSAVI dataset, which reflects changes in vegetation biomass and canopy cover, showed the best performance. The precision of the presented PRNN (RMSE = 0.85) in identification areas with a high potential of ignitability gives ground for the application of the proposed method in risk assessment and fuel treatment planning on WUI and adjoined urban areas.

**Keywords:** wildland–urban interface; vegetation ignitability; fire risk assessment; artificial neural network

## 1. Introduction

The concept of Wildland–Urban Interface (WUI) is a transition zone between the natural landscape and the build-up environment, officially proposed in February 1987 by the U.S. Department of Agriculture [1]. The basis for allocating territories to WUI with a specific fire management approach was reasoned by evidence that protecting structures from wildland fires is challenging, and human-caused fire ignitions are the most common, which became the basis for a specific legislative framework for WUI management [2]. Further, considering that anthropogenic factors increase the risk of a wildfire [3], the management of WUI decided to create buffer (sanitary) zones and fuel breaks to protect urban territories from fire [4]. A particular concern was that compact city planning is more resistant to fire [5] and that urban areas were not considered in fire management. The concept of non-flammable cities has worked for a long time until the climate changes observed in recent times led to an increase in the frequency and intensity of fire weather [6].

Nevertheless, wildfires that hit the city have become frequent in the last decade. Among numerous examples, wildfires have been reported in Greece (Athens, 2009 and 2015, Thasos, 2016, Mati, 2018), France (Marseilles, 2009 and 2016), Spain (Javea, 2012 and 2016), Italy (Palermo, 2022), Israel, 2016 and 2021, and the United States (California, 2018).

The catastrophic consequences of the wildfire spread in cities have made professional societies reconsider the existing approach to fire management and include urban vegetation adjoin to WUI in the risk assessments system [7].

Fire spread in forests and WUI is a combination of two main strategies: direct propagation from adjacent vegetation and spotting fire through ember attacks [8,9]. The specificity of urban areas limits direct propagation due to interspersing vegetation with fire-resistant structures, while firebrands are the primary fire spread strategy on the built-in part of the WUI [10–12]. In laboratory studies where fuel is subjected to contact with a lightning source, e.g., firebrands, the ignitability is defined as 100% due to the experiment conditions excluding external limiting factors, while the time of ignition and flaming duration vary [13,14]. Studies that have assessed actual wildfires, however, show that not all firebrands drive new ignitions; rather, fire spotting propagation depends on the meteorological conditions, the amount of fine fuel, and species composition on the specific patches [15]. Thus, the ability to identify areas with high ignition probability allows a better estimate fire connectivity network in the specific area, which in turn supports risk assessment and fuel treatment planning [16,17]. Although the probability of ignition depends on various environmental factors, there now exists a wealth of evidence that the main factor affecting the likelihood of the fire from firebrands is vegetation and its characteristics: moisture content, biomass amount, and biofuel type [18–20].

The three main approaches used to evaluate fire risks in vegetated areas are biophysical models, statistical models, and fire behavior models [21]

- Biophysical models estimate fire risk based on the scientifically validated weights of terrain physical parameters: vegetation, elevation, slope and aspect, roads, and settlements [22,23];
- Statistical models use GIS-based historical summaries to estimate the correlation between fire-affecting parameters and observed fire frequency at specific locations [24];
- Fire behavior models use mathematical models that predict fire spread based on biophysical parameters that simulate a fire dynamic in particular conditions [25]

Currently, the most advanced methodologies consider continuous risk assessment, learning from past events, and using dedicated techniques to process relevant data, support decisions, and enable risk management. Recent studies, therefore, propose a risk assessment approach based on machine learning [26,27]. Practice shows a good performance of various machine learning methods: support vector machines, decision trees, random forests, artificial neural networks (ANN), and k-nearest neighbors [28–30]. The advantage of ANN for fire studies is the ability to solve complex non-linear relationships between multiple inputs and the probability of ignition that allows for achieving predictive accuracy higher than in traditional statistical approaches [31]. ANNs are already implemented to predict forest fire probability based on common parameters for biophysical models. However, when considering the accuracy of estimated relationships to predict fire risks, both approaches have comparable difficulties in practical application due to spatial and temporal site specifics.

Although the vegetation characteristics are typical for fire risk, the specific dataset of vegetation parameters varies among studies, e.g., tree height, canopy cover, and vegetation type. The idea for monitoring vegetation state by spectral data obtained from remote sensing satellite missions appeared in 1970 [32]. Firstly, the studies were focused on the direct effect of plan biophysical properties, e.g., chlorophyll content, green biomass, and leaf area index, on vegetation reflectance in the different spectral ranges. As a result, a dozen vegetation indices (Vis) and statistical models were proposed for the estimation of vegetation characteristics. Further, the implementation of modern technologies and time-series spectral data allows for the estimation of indirect parameters such as plant phenology and forest overstorey fuel attributes and supports fuel model classification [33]. VIs allow for the detection of forest degradation [34], discriminating vegetation covers [35], and mapping vegetation according to the fuel type [36]. In practice, fire risk assessment uses datasets of time-series VIs for better fuel classification performance [37,38].

The present study focuses on supporting fire risk assessment on WUI and adjoined urban areas with the ANN application, which estimates the probability of ignition based on temporal VIs behavior. The study is conducted on empirical knowledge of Haifa's 2016 wildfire; we test the hypothesis that time series multispectral images provide sufficient information to predict vegetation ignitability using a pattern recognition neural network (PRNN).

## 2. Materials and Methods

### 2.1. Case Study

Haifa is a coastal Mediterranean city in Israel on Mt. Carmel (32°48′56″N, 34°59′21″E). The local topography includes steep mountain slopes and dry riverbeds (wadis) that frame the sprawling city with "green fingers", which leads to a considerable length of the WUI [39]. The urban area includes native greenery, flora, and many planted trees [40]. Within the city, the vegetation consists primarily of decorative plantings, low-growing trees, conifers, and maquis shrubland: *Pinus halepensis*, *Quercus* spp., *Quercus calliprinos*, *Ceratonia siliqua*, *Pistacia* spp., *Pistacia letiscus*, *Cistus salviifolius*, *Cistus criticus*, *Sarcopoterium spinosum*, *Calicotome villosa*, *Genista fasselata Decne* [41–43]. In the study region, the vegetation tends to be extremely flammable because of its short time-to-ignition and long flame duration. Previous ecological studies also indicate extensive connectivity among open spaces in Haifa; backyards and other urban in-between areas complement the semi-natural landscape and ensure wildlife movement between habitat patches [44].

Like the Mediterranean region, Haifa is prone to fires due to the "Fire Bioclimates" climate, characterized by dry and hot summers and wet and mild winters [45]. Easterly winds from deserts called "Sharav" aggravate the fire situation, which intensifies in the transitional seasons and brings high temperatures of nearly 40 °C and low humidity below 30% [46]. Wildfires in the areas surrounding Haifa are frequent and well-studied in the context of fire management; however, likely due to the non-flammable cities concept [39,47], urban areas are often excluded.

The wildfire considered in this study occurred in Haifa on 24 November 2016. Meteorological stations reported low humidity above 30% and strong southwest wind 10–15 m s$^{-1}$. The start of the fire was the ignition of wildland adjoined to the urban development [48]. Bypassing non-flammable constructions and burning nearly 9 ha of vegetation, the wildfire rapidly crossed the city in the first hour (Figure 1). The surrounding wildlands' total burned area was 120 ha [49]. According to the assessment, the total damage and loss amounted to 130,000,000 USD [50].

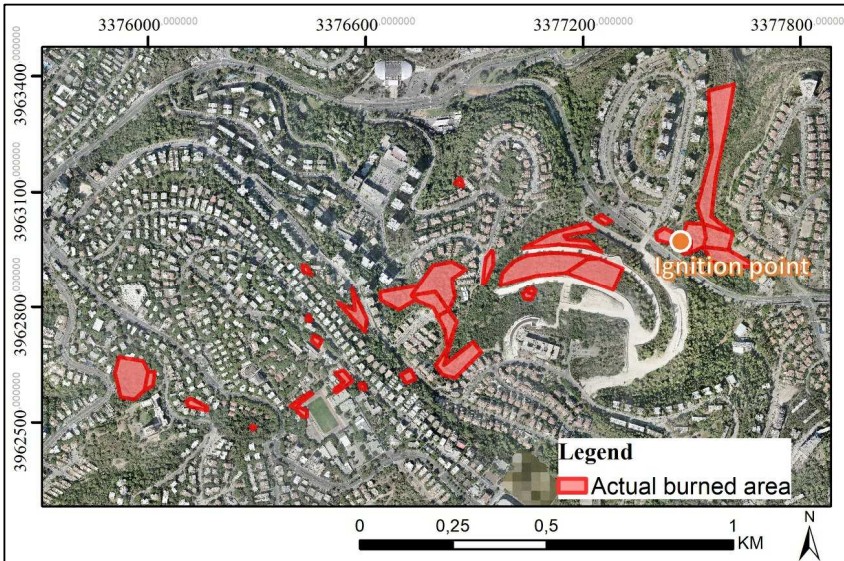

**Figure 1.** Map of the actual burned area at the first hour of the wildfire event in Haifa on 24 November 2016 [48].

*2.2. Data Collection and Pre-Processing*

The source for vegetation data collection is the LANDSAT 8 satellite which consists of two sensors: Operational Land Imager and Thermal Infrared Sensor. In the presented study, we use data from Operational Land Imager 8 (OLI8), which consists of 9 bands. Band 1 and band 9 are supporting bands for image correction according to environmental conditions (atmosphere and clouds). Bands 2–7 present visible and infrared spectral data. Bands 2–6 have spatial resolution of 30 m, and band 7 has resolution of 60 m. Band 8 is panchromatic channel proposed for data fusion and improvement of spatial resolution of the spectral bands to 15 m. We acquired OLI8 images with radiometric and geometric correction (Level-2 Data Product) during 2014–2016, as provided by the United States Geological Survey (USGS). Images have undergone atmospheric correction by the FLAASH algorithm [51] and Gram–Schmidt pan-sharpening [52] in the ENVI environment (L3Harris Technologies, Exelis Inc., Broomfield, CO, USA). Images with cloudiness of less than 5% on the region of interest were selected from the obtained data set. To reach a better spatial resolution that is important for urban studies, the collection of selected OLI8 images was downscaled to achieve 1 m resolution and to reconstruct pre-fire vegetation conditions [53]. The downscaling method is based on machine learning technique that estimates spatial distribution of vegetation from low-resolution spectra data by discovering dependencies between 1 m resolution aerial imagery and 15 m resolution satellite-acquired pixels.

The reference output data to train the proposed PRNN was a map of the actual burned area at the first hour of the wildfire event in Haifa on 24 November 2016, reconstructed based on crowd knowledge and firefighters' data by Polinova [48].

*2.3. Pattern Recognition Neural Network*

2.3.1. Neural Network Design

PRNN with the maximum likelihood principle was designed to estimate the relationship between temporal VIs and the probability of ignition on WUI and adjoined urban areas (Figure 2). Neural Network toolbox developed in MATLAB2020b environment (The MathWorks, Inc., Natick, MA, USA) that provides ready-use modules was chosen for PRNN designing. The backpropagation method with the sigmoid activation function was used for PRNN training. This approach allowed for the optimization of the weights and minimized a combination of squared errors so that the neural network learned how to estimate relationships between inputs and outputs correctly. To provide random data distribution for test-training-validation and to generalize the network by determining the correct combination of squared errors and weights, Bayesian regularization that eliminates the need for lengthy cross-validation process was applied for the designed PRNN. The number of hidden layers of PRNN was defined as 10 for all input datasets. The PRNN input data is temporal VIs behavior maps, and output data was classified into 'Ignited' and 'Not ignited' pixels.

2.3.2. PRNN Input

The collected and downscaled OLI8 images were processed to produce VIs maps. The VIs considered in the study were selected according to their relevance to the main fire-related vegetation characteristics: canopy cover, leaf area index, biomass amount, and moisture (Table 1). In total, six indices were considered in the study: Enhanced Vegetation Index (EVI), Normalized Difference Vegetation Index (NDVI), Modified Simple Ratio (MSR), Modified Soil Adjusted Vegetation Index (MSAVI), Transformed Difference Vegetation Index (TDVI), Normalized Multi-band Drought Index (NMDI).

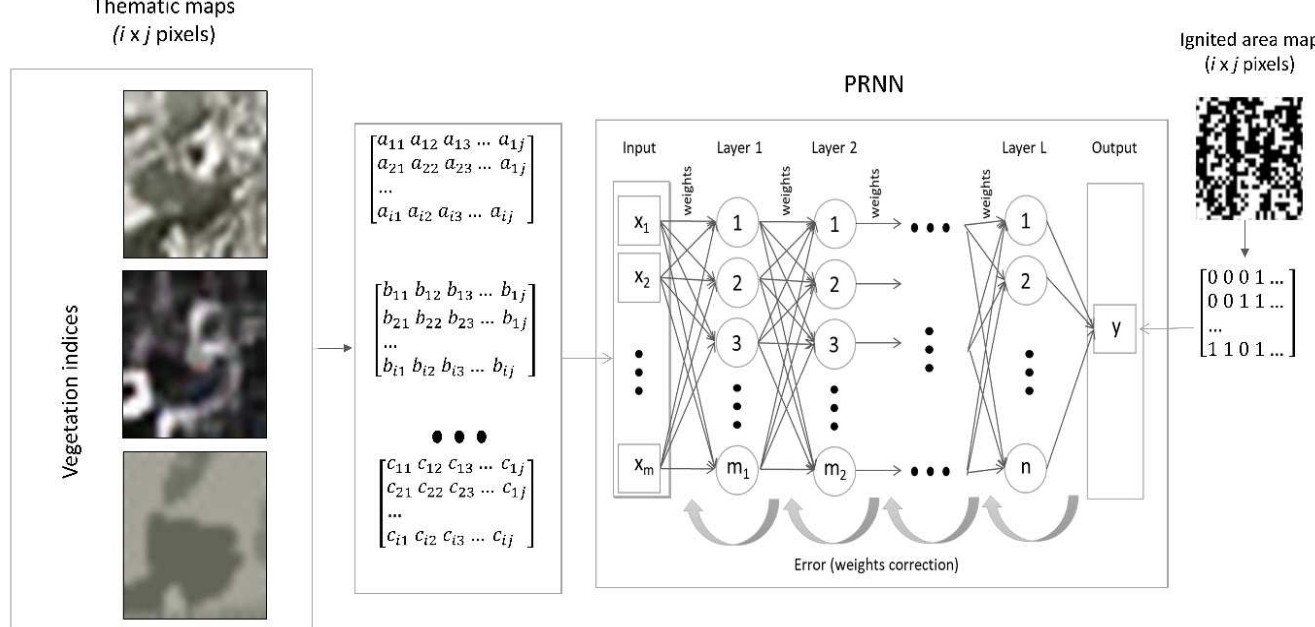

**Figure 2.** PRNN for estimating the relationship between temporal VIs behavior and probability of ignition.

**Table 1.** VIs considered in the study.

| VIs | Equation | Related Vegetation Characteristics | Reference |
|---|---|---|---|
| EVI | $2.5 * \frac{(NIR - Red)}{(NIR + 6 * Red - 7.5 * Blue + 1)}$ | Biomass, canopy cover | [54,55] |
| NDVI | $\frac{NIR - Red}{NIR + Red}$ | Biomass, tree productivity, leaf area index | [56–58] |
| MSR | $\frac{\left(\frac{NIR}{RED}\right) - 1}{\left(\sqrt{\frac{NIR}{Red}}\right) + 1}$ | Leaf area index, fraction of photosynthetically active radiation, biomass | [59,60] |
| MSAVI | $\frac{2 * NIR + 1 - \sqrt{(2 * NIr + 1)^2 - 8(NIR - Red)}}{2}$ | Biomass, canopy cover | [60–62] |
| TDVI | $1.5 * \left[\frac{(NIR - Red)}{\sqrt{NIR^2 + Red + 0.5}}\right]$ | Canopy cover | [63] |
| NMDI | $\frac{NIR - (SWIR1 - SWIR2)}{NIR + (SWIR1 - SWIR2)}$ | Vegetation moisture | [64] |

The PRNN input dataset had several configurations: all indices for all considered dates, each index for all dates, and a set of indices at the specific dates chosen based on the statistical analysis named the 'PCA dataset'. Factor analysis with the principal component method was used to reduce the number of variables by a multivariate technique that analyzes a matrix of numerous inter-correlated quantitative dependent variables [65,66]. 'PCA dataset' was proposed to evaluate factor-based combinations (different VIs at different dates) for better ignitability prediction.

### 2.3.3. PRNN Output

The map of the actual burned area at the first hour of the wildfire event was used to prepare the common output dataset for PRNNs with various inputs. Pixels matching the actual burned area at the first hour were marked to class 'Ignited'; pixels near the burned patches and staying resistant to the fire were mapped to the class 'Not ignited'. In total, 1500 pixels of each class were selected for the neural network training.

### 2.3.4. RNN Accuracy Measures

Trained PRNNs with the best performance of estimated ignition probability were subjected to accuracy assessment. The precision of the PRNNs was assessed by the percentage of pixels within the boundaries of the actual ignited area that is classified as 'Ignited' (i.e., with a probability threshold of 0.5). Pixels with an ignition probability of more than 0.5 are mapped together with polygons of the actual ignited area for visual inspection [67]. In addition, the quantitative assessment of accuracy was performed by calculating the average value and standard deviation of ignition probability, estimated by PRNNs in pixels falling into the actual ignited area.

## 3. Results

### 3.1. Data Preparation

After sorting satellite images and excluding images with a high level of cloudiness, the result was fourteen OLI8 images captured from November 2014 to October 2016. Due to the cloudiness, two large gaps occurred in the temporal data sequence; data was absent for the period from May to October 2015 and from March to July 2016. The resulting images were used to produce six VI's temporal maps—one temporal map contained all declared indices named 'All indices'.

The factor analysis was performed to reduce the number of indices from the 'All indices' dataset. The resulting two first principal components produced by the factor analysis explained 81.8% of the variance: 60.3% explained by the first component, and 21.5% explained by the second component. The strength of correlation between considered VIs and the first two principal components was used to select 39 VIs obtained at the specific dates and introduced in a new map named 'PCA dataset'.

### 3.2. PRNN Training

The predefined samples of pixels from the input maps were introduced in PRNN for training. The results of the confusion matrix for each prepared input dataset are presented in Table 2. All of the eight considered datasets showed good performance in identifying both ignited and not ignited pixels. When considering the prediction accuracy in each individual group—training, test, validation—we see a chaotic distribution of estimated accuracy caused by the random dividing of data into groups supported by Bayesian regularization. Therefore, to assess the PRNN performance regarding the input dataset, the total accuracy considering together results of training, test and validation was chosen as a key indicator. The lowest prediction accuracy was observed in the NMDI data set: 85.9% for ignited pixels and 87.5 for not ignited pixels. The 'All indices' dataset' accuracy was also relatively low—92.9% for ignited and 88.4% for not ignited. One more dataset with relatively low precision was PRNN trained on the NDVI dataset: 91.3% for ignited pixels and 83% for not ignited. The accuracy of PRNN trained by the TDVI dataset was 90.9% for ignited and 89.1% for not ignited pixels. The PRNN trained by the MSR dataset predicted ignited areas with an accuracy of 95% and not ignited with an accuracy of 86%. The neural network trained on the 'PCA dataset' and EVI had similar results in accuracy: 96.3% for ignited pixels in both datasets, 90.6% in the 'PCA dataset', and 87% in the EVI dataset for not ignited pixels. PRNN trained by the MSAVI dataset showed the best performance: 96.4% for ignited and 90.6% for not ignited areas. The best performance of trained PRNN was observed in MSAVI, EVI, and 'PCA dataset'.

### 3.3. PRNN Accuracy Assessment

Based on the training results, three ignition probability maps were reconstructed using PRNNs with the best performance: the EVI, MSAVI, and PCA datasets. A total of 94,043 pixels from the reconstructed PRNN map fell within the boundaries of the actual ignited area. Among them, a probability of ignition of more than 0.5 was 77.8% of the pixels estimated by PRNN trained on the EVI dataset, 85.6% of the pixels estimated by PRNN

trained on the MSAVI dataset, and 83.6% of the pixels estimated by PRNN trained on PCA dataset (Figure 3).

**Table 2.** The results of the Confusion Matrix of trained PRNN.

| Dataset | | Correspondence of Predicted and Actual Targets (%) | | | |
| --- | --- | --- | --- | --- | --- |
| | | Training | Validation | Test | Total (Training, Validation, and Test) |
| EVI | Ignited | 96.8 | 100 | 90 | 96.3 |
| | Not ignited | 82.4 | 93.8 | 100 | 87 |
| NDVI | Ignited | 90.6 | 100 | 86.7 | 91.3 |
| | Not ignited | 80.6 | 86.7 | 92.3 | 83 |
| MSR | Ignited | 96.3 | 90 | 93.8 | 95 |
| | Not ignited | 89 | 76.9 | 78.6 | 86 |
| MSAVI | Ignited | 96.9 | 90 | 100 | 96.4 |
| | Not ignited | 92.5 | 87.5 | 84.6 | 90.6 |
| TDVI | Ignited | 90.3 | 84.6 | 100 | 90.9 |
| | Not ignited | 91.5 | 77.8 | 93.3 | 89.1 |
| NMDI | Ignited | 84.5 | 88.9 | 91.7 | 85.9 |
| | Not ignited | 88.1 | 85.7 | 86.7 | 87.5 |
| All indices | Ignited | 94.1 | 90 | 85.7 | 92.9 |
| | Not ignited | 87.3 | 87.5 | 93.8 | 88.4 |
| PCA dataset | Ignited | 95.1 | 100 | 100 | 96.3 |
| | Not ignited | 84.5 | 100 | 92.9 | 87.9 |

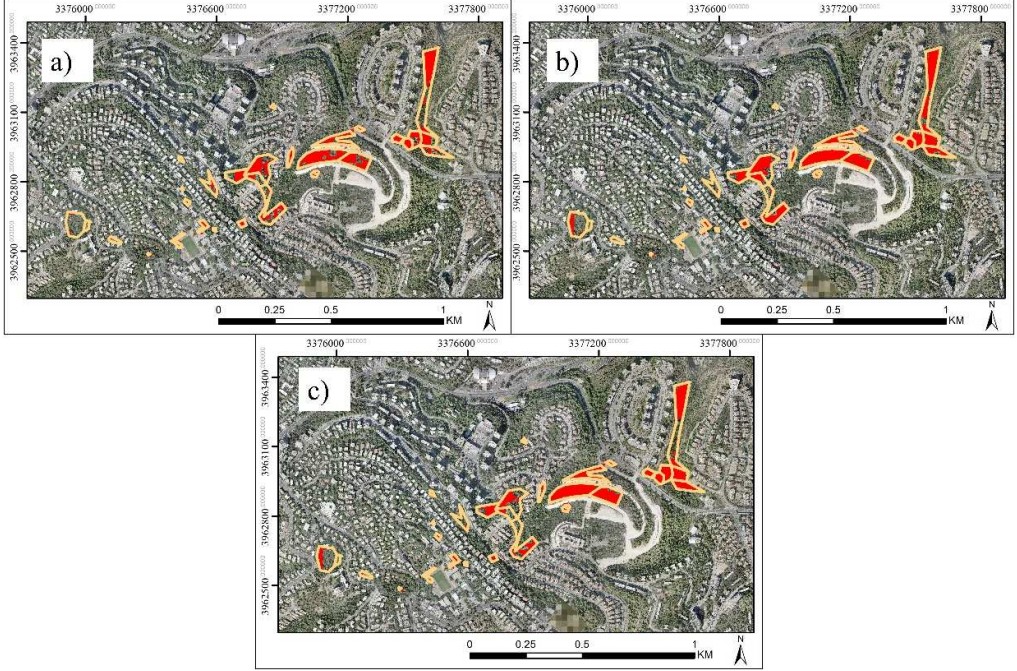

**Figure 3.** Maps of actual ignited area (orange line) via estimated ignited area with a probability more than 0.5 (red color): (**a**) by PRNN trained on EVI dataset; (**b**) by PRNN trained on MSAVI dataset; (**c**) by PRNN trained on 'PCA dataset'.

According to the methodology, the quantitative assessment of the predictive accuracy provided by the trained PRNNs with the best performance was calculated by the average value and standard deviation of ignition probability in pixels falling into the ignited area. Due to the ability of MSAVI to separate between vegetation and soil and sensitivity to plant dryness and phenology [68], the best precision for detecting ignited areas were observed in the map created by PRNN trained on this index: the average value of ignition probability was 80.21%, and the standard deviation was 24.9%. The PRNN trained on the PCA dataset also showed high precision in estimating ignition probability in the considered area: the average value was 80.06%, and the standard deviation was 26.28%. The lowest accuracy in prediction ignited areas was shown by PRNN trained on the EVI dataset: the average value was 73.57%, and the standard deviation was 31.17%.

## 4. Discussion

By utilizing biophysical vegetation characteristics expressed by VIs and calculated from OLS8 time series multispectral imagery, we effectively identified and mapped areas with a high-level probability of ignition within the WUI and adjoined urban areas. Whereas other methods based on the temporal deviation of vegetation characteristics obtain an RMSE $\approx$ 0.6–0.80 [69–71], the method used in this study was based on vegetation behavior patterns expressed by VIs and estimated to be more accurate (RMSE $\approx$ 0.85). It is important to note that obtained results, similar to other studies i.e., [69–71], refer to the informativeness of the input dataset and suitability of analytical technique, while the ability of generalization correlative approaches for fire risk assessment is still challenging [72]. The precision of the vegetation-only method presented in this paper is comparable to statistical approaches which consider complex environmental data [73–75] due to the advantage of machine learning techniques over traditional analyses [76,77].

The prediction accuracy of ignition probability by neural networks trained on complex environmental parameters depends highly on the number and informativeness of variables: the precision is directly correlated with the completeness of the input data [78]. The analysis of these networks highlights that among the considered environmental parameters, fuel moisture and the amount of precipitation are the main factors for ignitability prediction [79–81]. As noted in our introduction, vegetation is the main factor predicting ignitability and adds information about landscape and weather to increase the accuracy of the neural network analysis by a few percent [31]. The increased accuracy is because the information on vegetation state dynamics allows for obtaining indirect information about precipitation and anthropogenic activity by the plant growth rate [64,82,83].

The study area investigated in this paper represents a variety of vegetation types under different water regimes, including trees, shrubs, grass, and ornamental plants. The advantage of this work is the ability to predict the ignitability in a diversity of vegetation species with different moisture contents typical to urban areas, which has previously been a challenge for many researchers [74,84,85]. In contrast to the generally accepted approach of live fuel moisture content analysis as the primary ignitability estimator [86,87], spectral remote sending data and VIs, in particular, allow monitoring phenological status as relevant drivers of leaf biomass and moisture contents [88].

The feature of multispectral satellite systems such as OLS8 is collected by spectral signals together with the biochemical and physiological characteristics of vegetation [89]. The advantage of this feature for fire risk assessment is the ability to capture the changes both in water content [90] and phenology [91], which allows for the estimation of fuel flammability and supports ignitability prediction [13,14,92]. In the conducted study, the most appropriate VI for vegetation ignitability prediction on WUI and adjoint urban areas was MSAVI. Correlating with green biomass and vegetation cover, MSAVI makes the index a powerful tool for estimating vegetation vitality [93]. In recently published works, MSAVI has been established to predict land use and land cover classes such as native forest, shrublands, grassland, and vegetation adjoint to the built-up areas [94,95].

The high predictive accuracy of MSAVI observed in this study exceeded the results obtained with the input data configuration based on PCA analysis, indicating that the informativeness of this index in fire risk assessment is underestimated and has the potential for ignitability mapping. While NDVI is wildly used for fire risk assessment in vegetation [71], PRNN trained on the NDVI dataset is not among the top three methods in terms of accuracy due to the low sensitivity of the index to vegetation moisture in shrubs and trees [96].

The precision of the presented PRNN gives ground for applying this approach to estimate vegetation ignitability and can be implemented in fire risk assessment as input data that describes fuel [97,98]. MSAVI reflects vegetation characteristics relevant to flammability and, together with other environmental data such as topography and climate, can support fire management and decision-making on WUI and adjoined urban areas [92,99].

## 5. Conclusions

The present study proposes to support fire risk assessment on WUI and adjoined urban areas by estimating the probability of vegetation ignition by ANN. The PRNN was designed to predict ignitability based on temporal VIs behavior and assess its performance in comparison to the actual ignited area observed in a wildfire that occurred in Haifa, Israel, in 2016. The results of the study confirm that time series multispectral images provide sufficient information to classify vegetation according to its probability of ignition. Among the considered indices, the best performance in identifying areas with a high potential of ignitability was MSAVI, which reflects changes in vegetation biomass and canopy cover. The precision of more than 85% of the presented PRNN gives ground for applying this approach to assess vegetation ignitability and to support fire management and decision-making on WUI and adjoined urban areas.

**Author Contributions:** Conceptualization, M.M.; methodology, M.M.; software, M.M.; validation, M.M.; formal analysis, M.M.; investigation, M.M.; resources, M.M.; data curation, M.M.; writing—original draft preparation, M.M.; writing—review and editing, H.K., A.B. and L.W.; visualization, M.M.; supervision, H.K., A.B. and L.W.; project administration, A.B.; funding acquisition, A.B. All authors have read and agreed to the published version of the manuscript.

**Funding:** The authors declare that no funds, grants, or other support were received during the preparation of this manuscript.

**Institutional Review Board Statement:** Not applicable.

**Informed Consent Statement:** Not applicable.

**Data Availability Statement:** Not applicable.

**Acknowledgments:** Outstanding Doctoral Program "IDIT" at the Faculty of Social Science University of Haifa, Israel. We gratefully acknowledge the European Cooperation in Science and Technology (COST) action CA18135 "FIRElinks" (Fire in the Earth System: Science & Society) and CA16219 "HARMONIOUS" (Harmonization of UAS techniques for agricultural and natural ecosystems monitoring) for technical support.

**Conflicts of Interest:** The authors declare no conflict of interest.

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
