# Peer review of "Fire Risk Assessment on Wildland–Urban Interface and Adjoined Urban Areas: Estimation Vegetation Ignitability by Artificial Neural Network"

_fire, doi:10.3390/fire5060184_

Round 1
Reviewer 1 Report
General comments
I found this manuscript very interesting. Neural networks and other machine learning approaches clearly have great potential for predicting various aspects of fire risk and behaviour. However, I felt that in its present form the manuscript is somewhat over-confident about the results presented, to the extent that the text like a sales pitch in places.
The study only considers one fire in one study area, so while the results are encouraging and should motivate further work, they don’t provide any basis for judging how well the approach will generalize to future fires in this region, let alone to fires in other, substantially different regions. I’m aware that some theoretical papers support the generalizability of deep learning NN models (e.g. reference [3] below) but I’m not sure whether such an argument could be applied to the relatively simple NN model presented here. For this reason I would like to see the manuscript revised to present the results more in the spirit of a useful and interesting demonstration exercise with suggestions for further work, and with the over-confident statements about the superiority of this particular approach compared to other modelling methods removed.
Specific comments
Line 34-35: …that protecting structures from wildland fires is challenging, and human-caused fire ignitions are [2]
Incomplete sentence?
Line 40: …and practice confirmed the legality of its existence.
What does this mean?
Line 48-50: In contrast to fires in forest areas, where the major damage is associated with the amount of burnt fuel, urban wildfires are dangerous because they rapidly spread through vegetation patches by spotting fire ignition [7].
Not sure exactly what this sentence means. It seems to imply that spotting is not important in forest fires, or that it is more important in urban fire spread. This is incorrect, e.g. please see Storey et al 2020 (reference [1] below). Spotting is very important in many forest fires, while direct propagation from adjacent vegetation or buildings can be a major pathway in urban fires (reference [2] below).
Line 52: According to laboratory experiments, firebrands always cause vegetation ignition [10].
This sentence is worded as a general fact, but the cited reference only considered seven species grown as hedge plants in France.
Lines 64-69. The separation of “biophysical models” and “statistical models” seems very odd. Much of the statistical modelling of fire risk, rate of spread etc. is based on biophysical predictors. Perhaps it would be better to characterize approaches as either correlative (e.g. GIS-based historical summaries and/or statistical modelling) or process-based (fire simulation models).
Lines 77-79: The advantage of ANN for fire studies is the ability to solve complex non-linear relationships between multiple inputs and the probability of ignition.
This is true, but is it more difficult to infer causal processes from ANNs, random forests and similar approaches compared to parametric models? Also, does an ANN trained for one study area and time period always generalize easily to other study areas and time periods?
Lines 83-84: But since the physical properties of plants directly influence their reflectivity at different spectral ranges [28], vegetation indices (VIs) can describe the vegetation state.
The cited reference is very old (1970). Many important properties of vegetation cannot be reliably derived from spectral remote sensing data, e.g. vertical connectivity between ground, understorey and overstorey layers.
Line 131: The source for vegetation data collection is Operational Land Imager 8 (OLI8).
It would be useful to also state that this is instrument is on the LANDSAT 8 satellite and to specify the resolution of the original data (15m?)
Lines 138-139: …selected OLI8 images was downscaled to achieve 1m resolution and to reconstruct pre-fire vegetation conditions [48].
Please provide some brief details about the down-scaling (up-sampling) methods used rather than requiring the reader to consult the cited reference.
Lines 152-154: Bayesian regularization for PRNN was proposed to provide random data distribution for test-training-validation and to generalize the network by determining the correct combination of squared errors and weights
Does “proposed” mean that this is what the authors did? If so, this term will be unfamiliar to many readers so a few words of explanation and a reference would be helpful.
Lines 176-177: The strength of correlation between considered VIs and informative principal components defined the significance of the variable for prediction ignited areas.
Not quite sure I understand this correctly. Does it mean that principal components served as predictors in the ANN and then, for those that appeared to be informative, linear correlations were calculated with respect to individual vegetation indices? If so, what was the purpose in doing this? More generally, what was the purpose of using principal components as predictors rather than just the vegetation indices? This question might just reflect my lack of experience with neural networks, but I suspect that will also be true for many readers. Please also see the related comment below regarding lines 194-207.
Lines 187-188: …classified as ‘Ignited’ (i.e., with a probability threshold of 0.5)
Was there any consideration of using a threshold other than 0.5? Perhaps exploring other thresholds, taking into account the differing cost of mis-classification of ignited and unignited points, would be useful.
Lines 194-207: Data preparation
This text would be better placed in the Methods section. Related to my earlier comment, I am still unclear about the motivation for doing the PCA. I also worry that this approach to selecting predictors will not always be reliable. Correlations between an individual variable and the first two axes of the PCA indicate the degree to which that variable is similar to many others, but this does not necessarily mean it is a good choice as a predictor. There could be a variable that is poorly correlated with any others, and therefore also poorly correlated with the dominant PCA axes, but that would be a superior predictor. This is a well-recognized problem in regression analysis, e.g. see the discussion at https://win-vector.com/2016/05/16/pcr_part1_xonly/
Table 2. Line 224
I’m a little confused about several rows in the table where the accuracy for ‘Test’ data is 100% while the accuracy for ‘Training’ and ‘Validation’ data is quite a bit lower, e.g. EVI/Not-ignited and TDVI/Ignited. Perhaps a little more explanation in the accompanying text would help. Also, how were the data partitioned into test, training and validation data sets?
Line 240: As expected, the best precision for detecting ignited areas was observed in the map created by PRNN trained on the MSAVI dataset
Why was this expected?
Lines 251-254: Whereas other methods based on the temporal deviation of vegetation characteristics obtain an RMSE ≈ 0.6 - 0.80 [63-65], the methods used in this study were based on vegetation behavior patterns expressed by VIs and estimated to be more accurate (RMSE 0.85)
I think the wording here is somewhat over-confident. The accuracy results cited as a comparison are only from three studies. Moreover, this accuracy results for the present study is only for one fire and there it has not been shown that this model or the general method will generalize as successfully to other fires and other study areas.
Line 258: …informativity…
Is that a real word? Perhaps “informativeness”?
Lines 269-272: The advantage of this work is the high accuracy of predicting ignitability in a diversity of vegetation species with different moisture contents, which has previously been a challenge for many researchers
I think this statement risks over-stating the results and should be removed. As far as I can see, the authors present accuracy statistics for the whole study, i.e. aggregated over the various species and moisture contents present. There are no results differentiating accuracy for individual species, groups of species or moisture states.
Lines 273-276: …our results were achieved due to the multilateral consideration of vegetation state dynamics and acknowledgement of distinctive phenological status as relevant drivers of leaf biomass and moisture contents [81].
What does this mean?
References cited above (please note: I am not an author on any of these papers)
[1] Storey Michael A., Price Owen F., Sharples Jason J., Bradstock Ross A. (2020) Drivers of long-distance spotting during wildfires in south-eastern Australia. International Journal of Wildland Fire 29, 459-472. https://doi.org/10.1071/WF19124
[2] Penman Sandra H., Price Owen F., Penman Trent D., Bradstock Ross A. (2019) The role of defensible space on the likelihood of house impact from wildfires in forested landscapes of south eastern Australia. International Journal of Wildland Fire 28, 4-14. https://doi.org/10.1071/WF18046
[3] Mingard, Chris and Valle-Pérez, Guillermo and Skalse, Joar and Louis, Ard A. (2020) Is SGD a Bayesian sampler? Well, almost. https://doi.org/10.48550/arxiv.2006.15191
Author Response
The authors greatly appreciated your constructive and helpful comments/suggestions. We have included below a point-by-point response to the raised concerns.
We believe that the manuscript is significantly improved after this revision. Additionally, we have tracked the changes in the revised manuscript.
Line 34-35: …that protecting structures from wildland fires is challenging, and human-caused fire ignitions are [2]
Incomplete sentence?
My apologies for the technical mistake. The complete sentence is:
Line 34-36: “The basis for allocating territories to WUI with a specific fire management approach was reasoned by evidence that protecting structures from wildland fires is challenging, and human-caused fire ignitions are most common, became the basis for a specific legislative framework for WUI management [2].”
Line 40: …and practice confirmed the legality of its existence.
What does this mean?
The sentence was improved:
Line 40-43: “The concept of non-flammable cities has worked for a long time until the climate changes observed in recent times led to an increase in frequency and intensity of fire weather [6].
[6] Pörtner, H. O., Roberts, D. C., Adams, H., Adler, C., Aldunce, P., Ali, E., ... & Fischlin, A..Climate change 2022: Impacts, adaptation and vulnerability. IPCC Sixth Assessment Report 2022.
Line 48-50: In contrast to fires in forest areas, where the major damage is associated with the amount of burnt fuel, urban wildfires are dangerous because they rapidly spread through vegetation patches by spotting fire ignition [7].
Not sure exactly what this sentence means. It seems to imply that spotting is not important in forest fires, or that it is more important in urban fire spread. This is incorrect, e.g. please see Storey et al 2020 (reference [1] below). Spotting is very important in many forest fires, while direct propagation from adjacent vegetation or buildings can be a major pathway in urban fires (reference [2] below).
Thank you, I used recommended references to cover the topic in more detail ([9 – yours 2] refers to houses in forested landscape, [12] declare that spotting fire is the main strategy for fire spread through built-up environment):
Line 51-58 “Fire spread in forests and WUI is a combination of two main strategies: direct propagation from adjacent vegetation and spotting fire through ember attacks [8.9]. Specificity of urban areas limits direct propagation due to interspersing vegetation with fire-resistant structures, while firebrands are the primary fire spread strategy on the built-in part of the WUI [10,11,12].”
[8] Storey Michael A., Price Owen F., Sharples Jason J., Bradstock Ross A. (2020) Drivers of long-distance spotting during wildfires in south-eastern Australia. International Journal of Wildland Fire 2020, 29, 459-472.
[9] Penman Sandra H., Price Owen F., Penman Trent D., Bradstock Ross A. The role of defensible space on the likelihood of house impact from wildfires in forested landscapes of south eastern Australia. International Journal of Wildland Fire 2019, 28, 4-14.
Line 52: According to laboratory experiments, firebrands always cause vegetation ignition [10].
This sentence is worded as a general fact, but the cited reference only considered seven species grown as hedge plants in France.
There is a series of works by Ganteaume discussing fuel ignition and firebrands. I agree that reference I used before not fully fit the idea. I added description for clarification and reference to earlier paper [13]:
Line 58-62: “In laboratory studies where fuel subjected to contact with lightning source, e.g. firebrands, the ignitability is defined as 100% due to the experiment conditions excluding external limiting factors, while time of ignition and flaming duration are varying [13,14].”
[13] Ganteaume, A., Guijarro, M., Jappiot, M., Hernando, C., Lampin-Maillet, C., Pérez-Gorostiaga, P., & Vega, J. A.. Laboratory characterization of firebrands involved in spot fires. Annals of Forest Science 2011, 68(3), 531-541.
Lines 64-69. The separation of “biophysical models” and “statistical models” seems very odd. Much of the statistical modelling of fire risk, rate of spread etc. is based on biophysical predictors. Perhaps it would be better to characterize approaches as either correlative (e.g. GIS-based historical summaries and/or statistical modelling) or process-based (fire simulation models).
Thank you for the attention to this issue. I share your opinion that the biophysical and statistical models have similarities since use same correlative techniques and inputs. However, in practice the models have difference in methodology design and potential implementation of the results. I adapted original description to highlight it:
Line 73-80
“a) Biophysical models assess fire risk based on the scientifically validated weights of terrain physical parameters: vegetation, elevation, slope and aspect, roads, and settlements [22,23];
- b) Statistical models use GIS-based historical summaries to estimate the correlation between fire affecting parameters and observed fire frequency at specific location [24];”
Lines 77-79: The advantage of ANN for fire studies is the ability to solve complex non-linear relationships between multiple inputs and the probability of ignition.
This is true, but is it more difficult to infer causal processes from ANNs, random forests and similar approaches compared to parametric models? Also, does an ANN trained for one study area and time period always generalize easily to other study areas and time periods?
Thank you for highlighting the aspect of practical application of these methods. You referred to the common problems of correlative approaches for application in fire management due to sites specifics that limits development worldwide fire risk assessment system and requires from each country or region develop national programs. I agree that it is important to mention this in the article. The text was adjusted:
Line 88-96 “The advantage of ANN as one of machine learning methods for fire studies is the ability to solve complex non-linear relationships between multiple inputs and the probability of ignition, that allows to achieve predictive accuracy higher than in traditional statistical approaches [31]. ANNs are already implemented to predict forest fire probability based on common parameters for biophysical models. However, when considering accuracy of estimated relationships to predict fire risks, both approaches have comparable difficulties in practical application due to spatial and temporal sites specifics.
Lines 83-84: But since the physical properties of plants directly influence their reflectivity at different spectral ranges [28], vegetation indices (VIs) can describe the vegetation state.
The cited reference is very old (1970). Many important properties of vegetation cannot be reliably derived from spectral remote sensing data, e.g. vertical connectivity between ground, understorey and overstorey layers.
The topic was expanded to cover issues that you mentioned:
Line 99-108: “The idea for monitoring vegetation state by spectral data obtained from remote sensing satellite missions appeared in 1970 [32]. Firstly, the studies were focused on direct effect of plan biophysical properties e.g., chlorophyll content, green biomass and leaf are index on vegetation reflectance in different spectral range. As a result, a dozen vegetation indices (Vis) and statistical models were proposed for estimation vegetation characteristics. Further, implementation modern technologies and time-series spectral data allows to estimate indirect parameters such as plant phenology, forest overstorey fuel attributes and supports fuel models classification [33].”
[33] Gale, M. G., Cary, G. J., Van Dijk, A. I., & Yebra, M.. Forest fire fuel through the lens of remote sensing: Review of approaches, challenges and future directions in the remote sensing of biotic determinants of fire behaviour. Remote Sensing of Environment 2021, 255, 112282.
Line 131: The source for vegetation data collection is Operational Land Imager 8 (OLI8).
It would be useful to also state that this is instrument is on the LANDSAT 8 satellite and to specify the resolution of the original data (15m?)
Done:
Line 155-162: “The source for vegetation data collection is the LANDSAT 8 satellite that consists of two sensors: Operational Land Imager and Thermal Infrared Sensor. In the presented study we use data from Operational Land Imager 8 (OLI8) that consists of 9 bands. Band 1 and band 9 are supporting bands for image correction according to environmental conditions (atmosphere and clouds). Bands 2-7 presents visible and infrared spectral data. Bands 2-6 have spatial resolution 30m and band 7 with resolution 60m. Band 8 is panchromatic channel proposed for data fusion and improvement spatial resolution of the spectral bands to 15m.”
Lines 138-139: …selected OLI8 images was downscaled to achieve 1m resolution and to reconstruct pre-fire vegetation conditions [48].
Please provide some brief details about the down-scaling (up-sampling) methods used rather than requiring the reader to consult the cited reference.
Brief description was added:
Line 170-173: “The downscaling method is based on machine learning technique that estimates spatial distribution of vegetation from low-resolution spectra data by discovering dependencies between 1m resolution aerial imagery and 15m resolution satellite-acquired pixels.”
Lines 152-154: Bayesian regularization for PRNN was proposed to provide random data distribution for test-training-validation and to generalize the network by determining the correct combination of squared errors and weights
Does “proposed” mean that this is what the authors did? If so, this term will be unfamiliar to many readers so a few words of explanation and a reference would be helpful.
Done:
Line 189-195: “To provide random data distribution for test-training-validation and to generalize the network by determining the correct combination of squared errors and weights, Bayesian regularization that eliminates the need for lengthy cross-validation process was applied for the designed PRNN.”
Lines 176-177: The strength of correlation between considered VIs and informative principal components defined the significance of the variable for prediction ignited areas.
Not quite sure I understand this correctly. Does it mean that principal components served as predictors in the ANN and then, for those that appeared to be informative, linear correlations were calculated with respect to individual vegetation indices? If so, what was the purpose in doing this? More generally, what was the purpose of using principal components as predictors rather than just the vegetation indices? This question might just reflect my lack of experience with neural networks, but I suspect that will also be true for many readers. Please also see the related comment below regarding lines 194-207.
The description of PCA propose is improved:
Line 213-221: “Factor analysis with the principal component method was used to reduce the number of variables by a multivariate technique that analyzes a matrix of numerous intercorrelated quantitative dependent variables [65,66]. 'PCA dataset' was proposed to evaluate factor-based combinations (different VIs at different dates) for better ignitability prediction.”
Lines 187-188: …classified as ‘Ignited’ (i.e., with a probability threshold of 0.5)
Was there any consideration of using a threshold other than 0.5? Perhaps exploring other thresholds, taking into account the differing cost of mis-classification of ignited and unignited points, would be useful.
To avoid the heavy work of checking thresholds and to concentrate on the informativeness of the input data as the main study propose, the classification method into 2 classes was chosen: ignited and not ignited. So, the threshold of 0.5 is the point where both options are equally possible. While deviation from the 0.5 means apriory belonging to one of the classes.
Lines 194-207: Data preparation
This text would be better placed in the Methods section. Related to my earlier comment, I am still unclear about the motivation for doing the PCA. I also worry that this approach to selecting predictors will not always be reliable. Correlations between an individual variable and the first two axes of the PCA indicate the degree to which that variable is similar to many others, but this does not necessarily mean it is a good choice as a predictor. There could be a variable that is poorly correlated with any others, and therefore also poorly correlated with the dominant PCA axes, but that would be a superior predictor. This is a well-recognized problem in regression analysis, e.g. see the discussion at https://win-vector.com/2016/05/16/pcr_part1_xonly/
The explanation of PCA was improved in methodology. To avoid confusion the part was deleted (Line 246-247) “it saved sufficiently informative data to indicate ignited areas using multispectral images”
Table 2. Line 224
I’m a little confused about several rows in the table where the accuracy for ‘Test’ data is 100% while the accuracy for ‘Training’ and ‘Validation’ data is quite a bit lower, e.g. EVI/Not-ignited and TDVI/Ignited. Perhaps a little more explanation in the accompanying text would help. Also, how were the data partitioned into test, training and validation data sets?
The explanation was added (also in methodology – Line 189-195):
Line 256-261: “When considering the prediction accuracy in each individual group - training, test, validation - we see a chaotic distribution of estimated accuracy caused by the random dividing of data into groups supported by Bayesian regularization. Therefore, to assess the PRNN performance regarding to input dataset, the total accuracy considering together results of training, test and validation was chosen as a key indicator.”
Line 240: As expected, the best precision for detecting ignited areas was observed in the map created by PRNN trained on the MSAVI dataset
Why was this expected?
The sentence is clarified:
Line 289-291: “Due to the ability of MSAVI to separate between vegetation and soil and sensitivity to plant dryness and phenology [68], the best precision for detecting ignited areas was observed in the map created by PRNN trained on this index:”
[68] Guerra, F., Puig, H., & Chaume, R.The forest-savanna dynamics from multi-date Landsat-TM data in Sierra Parima, Venezuela. International Journal of Remote Sensing 1998, 19(11), 2061-2075.
Lines 251-254: Whereas other methods based on the temporal deviation of vegetation characteristics obtain an RMSE ≈ 0.6 - 0.80 [63-65], the methods used in this study were based on vegetation behavior patterns expressed by VIs and estimated to be more accurate (RMSE 0.85)
I think the wording here is somewhat over-confident. The accuracy results cited as a comparison are only from three studies. Moreover, this accuracy results for the present study is only for one fire and there it has not been shown that this model or the general method will generalize as successfully to other fires and other study areas.
To cover the topic that you mentioned we added the sentence:
Line 305-308: “It is important to note that obtained results, similar to other studies [i.e. 69-71], refers to informativeness of input dataset and suitability of analytical technique, while ability of generalization correlative approaches for fire risk assessment is still challenging [72].”
[72] Xie, L., Zhang, R., Zhan, J., Li, S., Shama, A., Zhan, R., ... & Wu, R. Wildfire Risk Assessment in Liangshan Prefecture, China Based on An Integration Machine Learning Algorithm. Remote Sensing 2022, 14(18), 4592.
We want to avoid glut of references in the paper. We ready to provide some additional examples of studies with the accuracy 0.6-0.8:
Wen, C., He, B., Quan, X., Liu, X., & Liu, X. (2018, July). Wildfire Risk Assessment Using Multi-Source Remote Sense Derived Variables. In IGARSS 2018-2018 IEEE International Geoscience and Remote Sensing Symposium (pp. 7644-7647). IEEE.
Chéret, V., Sampara, J. D. W., & Gay, M. (2007). Time series analysis of remote sensing to calculate and map operational indicators of wildfire risk. TOWARDS AN OPERATIONAL USE OF REMOTE SENSING IN FOREST FIRE MANAGEMENT, 143.
Michael, Y., Helman, D., Glickman, O., Gabay, D., Brenner, S., & Lensky, I. M. (2021). Forecasting fire risk with machine learning and dynamic information derived from satellite vegetation index time-series. Science of The Total Environment, 764, 142844.
Line 258: …informativity…
Is that a real word? Perhaps “informativeness”?
Done: Line 312
Lines 269-272: The advantage of this work is the high accuracy of predicting ignitability in a diversity of vegetation species with different moisture contents, which has previously been a challenge for many researchers
I think this statement risks over-stating the results and should be removed. As far as I can see, the authors present accuracy statistics for the whole study, i.e. aggregated over the various species and moisture contents present. There are no results differentiating accuracy for individual species, groups of species or moisture states.
The sentence was clarified:
Line 323-327: “The study area investigated in this paper represents a variety of vegetation types under different water regimes, including trees, shrubs, grass, and ornamental plants. The advantage of this work is ability to predict the ignitability in a diversity of vegetation species with different moisture contents typical to urban areas, which has previously been a challenge for many researchers [74,84,85].”
Lines 273-276: …our results were achieved due to the multilateral consideration of vegetation state dynamics and acknowledgement of distinctive phenological status as relevant drivers of leaf biomass and moisture contents [81].
What does this mean?
Rewritten:
Line 327-332: “In contrast to the generally accepted approach of live fuel moisture content analysis as the primary ignitability estimator [86,87], spectral remote sending data and VIs in particular allow to monitor phenological status as relevant drivers of leaf biomass and moisture contents [88].”
References cited above (please note: I am not an author on any of these papers)
[1] Storey Michael A., Price Owen F., Sharples Jason J., Bradstock Ross A. (2020) Drivers of long-distance spotting during wildfires in south-eastern Australia. International Journal of Wildland Fire 29, 459-472. https://doi.org/10.1071/WF19124
[2] Penman Sandra H., Price Owen F., Penman Trent D., Bradstock Ross A. (2019) The role of defensible space on the likelihood of house impact from wildfires in forested landscapes of south eastern Australia. International Journal of Wildland Fire 28, 4-14. https://doi.org/10.1071/WF18046
[3] Mingard, Chris and Valle-Pérez, Guillermo and Skalse, Joar and Louis, Ard A. (2020) Is SGD a Bayesian sampler? Well, almost. https://doi.org/10.48550/arxiv.2006.15191
Reviewer 2 Report
This is a great work as it employs new concepts such as machine vision, artificial neural network and image processing in wildfire research. I believe there is a huge potential in in advancing the wildfire science using these techniques.
Some minor points:
- How does The number of layers of PRNN effects for all input datasets?
- Regarding the losses, I would recommend to follow universal currency (USD);
- Consider it necessary to indicate in the article several arguments in favor of choosing the Matlab environment for implementing PRNN;
- The term "actual ignited ares" is used in the article. For greater certainty – the number of pixels and the accuracy of these boundaries is determined by the ability of the recording equipment? My question is related to the limits of applicability of this method, as well as the issue of increasing accuracy. Having an image that is higher in spatial resolution, can one get a more accurate forecast using PRNN? Is the dependence between resolution and accuracy more complex and has a number of factors? (With the current number of vegetation indices);
- The source for vegetation data collection was Operational Land Imager 8 (OLI8). Please clarify the technical specifications of this device, which was used for obtaining the images;
- The authors argue in conclusion about the possibility of using the data obtained in fire risk assessment as input data that describes fuel. It would be recommended to refer to specific methods/literature data in order to see applicability more clearly. This issue in a broader context is also related to the possibility of applicability of such data in different fire hazard forecasting methods, how easy it is to adjust the output data to one or another method widely used in the world.
Author Response
The authors greatly appreciated your constructive and helpful comments/suggestions. We have included below a point-by-point response to the raised concerns.
We believe that the manuscript is significantly improved after this revision. Additionally, we have tracked the changes in the revised manuscript.
- How does The number of layers of PRNN effects for all input datasets?
Application ANN in fire risk assessment is a relatively new topic. The presented study is focused on evaluation the input dataset for estimation ignitability of vegetation in the city. For this propose PRNN configuration following default hyper-parameters in MATLAB2020b. The number of layers was designed based on rule-of-thumb methods taking in account the data complexity.
I agree that the ANN precision depends on number of hidden layers (together with the effect of input/output neurons number) and this topic should be covered in the future. The dataset with highest prediction accuracy obtained in this study can be used as constant data for a new study dedicated to evaluation the effect of layers number
- Regarding the losses, I would recommend to follow universal currency (USD);
Done: Line 150 130,000,000USD
- Consider it necessary to indicate in the article several arguments in favor of choosing the Matlab environment for implementing PRNN;
In the presented work, the main focus is on wildfires, vegetation, and remote sensing. The advantage of MATLAB in this case is easy to use. We added it in the text:
Line 181-187: “Neural Network toolbox developed in MATLAB2020b environment (The MathWorks, Inc., Natick, Massachusetts, USA) that provides ready-use modules was chosen for PRNN designing. The backpropagation method with the sigmoid activation function was used for PRNN training.”
- The term "actual ignited ares" is used in the article. For greater certainty – the number of pixels and the accuracy of these boundaries is determined by the ability of the recording equipment? My question is related to the limits of applicability of this method, as well as the issue of increasing accuracy. Having an image that is higher in spatial resolution, can one get a more accurate forecast using PRNN? Is the dependence between resolution and accuracy more complex and has a number of factors? (With the current number of vegetation indices);
The traditional resampling methods known in image processing studies has limitations in estimation fire related vegetation properties. Since vegetation is a part of ecosystem, it should be considered in a single vein with the ecological scaling approach: individual, population, community. The presented study was conducted on urban scale and the spatial resolution was sufficient to capture information on individual level. Increasing in spatial resolution moves the study on leaf scale where photosynthesis effect on spectra higher than phenology.
- The source for vegetation data collection was Operational Land Imager 8 (OLI8). Please clarify the technical specifications of this device, which was used for obtaining the images;
Done:
Line 155-162: “The source for vegetation data collection is the LANDSAT 8 satellite that consists of two sensors: Operational Land Imager and Thermal Infrared Sensor. In the presented study we use data from Operational Land Imager 8 (OLI8) that consists of 9 bands. Band 1 and band 9 are supporting bands for image correction according to environmental conditions (atmosphere and clouds). Bands 2-7 presents visible and infrared spectral data. Bands 2-6 have spatial resolution 30m and band 7 with resolution 60m. Band 8 is panchromatic channel proposed for data fusion and improvement spatial resolution of the spectral bands to 15m.”
- The authors argue in conclusion about the possibility of using the data obtained in fire risk assessment as input data that describes fuel. It would be recommended to refer to specific methods/literature data in order to see applicability more clearly. This issue in a broader context is also related to the possibility of applicability of such data in different fire hazard forecasting methods, how easy it is to adjust the output data to one or another method widely used in the world.
The references were added:
Line 350-355: “The precision of the presented PRNN gives ground for applying this approach to estimate vegetation ignitability and can be implemented in fire risk assessment as in-put data that describes fuel [98,99]. MSAVI reflects vegetation characteristics relevant to flammability and, together with other environmental data such as topography and climate, can support fire management and decision-making on WUI and adjoined urban areas [92,100].”
[98] Blackhall, M., & Raffaele, E. Flammability of Patagonian invaders and natives: When exotic plant species affect live fine fuel ignitability in wildland-urban interfaces. Landscape and urban planning 2019, 189, 1-10.
[99]Farahmand, A., Reager, J. T., Behrangi, A., Stavros, E. N., & Randerson, J. T. Using NASA satellite observations to map wildfire risk in the United States for allocation of fire management resources. In AGU Fall Meeting Abstracts 2017 (Vol. 2017, pp. NH21E-02).
[100]Galiana-Martin, L., Herrero, G., & Solana, J. A wildland–urban interface typology for forest fire risk management in Mediterranean areas. Landscape Research 2011, 36(2), 151-171.